# Lower Semen Quality Among Men in the Modern Era—Is There a Role for Diet and the Microbiome?

**DOI:** 10.3390/microorganisms13010147

**Published:** 2025-01-13

**Authors:** Kristina Magoutas, Sebastian Leathersich, Roger Hart, Demelza Ireland, Melanie Walls, Matthew Payne

**Affiliations:** 1Division of Obstetrics and Gynaecology, Medical School, University of Western Australia, Perth, WA 6009, Australia; kristina.magoutas@research.uwa.edu.au (K.M.); roger.hart@uwa.edu.au (R.H.); 2Fertility Specialists of Western Australia (City Fertility), Perth, WA 6153, Australia; sebastian.leathersich@gmail.com; 3Dexeus Fertility, Department of Obstetrics, Gynecology and Reproductive Medicine, Hospital Universitari Dexeus, 08028 Barcelona, Spain; 4Department of Obstetrics, Gynecology and Reproductive Medicine, Universitat de Barcelona, 08007 Barcelona, Spain; 5School of Biomedical Sciences, University of Western Australia, Perth, WA 6009, Australia; demelza.ireland@uwa.edu.au; 6Concept Fertility, Perth, WA 6008, Australia; melanie.walls@conceptfertility.com.au

**Keywords:** semen quality, sperm, male infertility, diet, microbiome

## Abstract

The prevalence of infertility is increasing worldwide; poor nutrition, increased sedentary lifestyles, obesity, stress, endocrine-disrupting chemicals, and advanced age of childbearing may contribute to the disruption of ovulation and influence oocyte and sperm quality and overall reproductive health. Historically, infertility has been primarily attributed to female factors, neglecting the importance of male fertility; this has resulted in an incomplete understanding of reproductive health. Male factors account for 40–50% of infertility cases. In half of these cases, the proximal cause for male infertility is unknown. Sperm contributes half of the nuclear DNA to the embryo, and its quality is known to impact fertilisation rates, embryo quality, pregnancy rates, risk of spontaneous miscarriage, de novo autosomal-dominant conditions, psychiatric and neurodevelopment conditions, and childhood diseases. Recent studies have suggested that both the microenvironment of the testes and diet quality may play an important role in fertility; however, there is limited research on the combination of these factors. This review summarises current known causes of male infertility and then focuses on the potential roles for diet and the seminal microbiome. Future research in this area will inform dietary interventions and health advice for men with poor semen quality, potentially alleviating the need for costly and invasive assisted reproduction treatments and allowing men to take an active role in the fertility conversation which has historically focussed on women individually.

## 1. Introduction

Infertility is the inability to achieve a clinical pregnancy within 12 months of regular unprotected sexual intercourse [1]. It affects up to 15% of couples worldwide, with 40–50% of these cases being attributable to male factors [1]. Male fertility is defined as the ability to contribute to conception through the production of healthy sperm, delivery of the sperm into the female reproductive tract, and the ability of the sperm to fertilise an ovum successfully. Causes of male infertility may be congenital or acquired and may be pre-testicular, testicular, or post-testicular. Approximately half of male infertility cases are classified as idiopathic [2]. Age, smoking, alcohol consumption, and environmental factors such as endocrine-disrupting chemicals have been shown to negatively impact male fertility [3,4]. Epidemiological studies in human and animal models have shown that endocrine disruption can lead to the testicular dysgenesis syndrome, compromising cryptorchidism, hypospadias, impaired spermatogenesis, and an increased risk of testicular cancer. Despite this, compounds such as phthalates continue to be used as a plasticiser for packaging, medical devices, and pharmaceuticals [5,6,7]. Along with a range of other factors, these endocrine-disrupting chemical exposures are believed to be contributing to the decline in sperm concentration seen globally, over the last 50 years [8,9,10,11,12,13].

Assisted reproductive technology (ART) provides a treatment option for various causes of infertility, including male factor and unexplained infertility. While ART births in Australia have increased by 55% between 2011 and 2021, live birth rates per cycle improved only slightly [14]. ART can overcome some aspects of male factor infertility, but sperm quality remains imperative for fertilisation, embryo development and offspring health later in life [15]. Though ART has supported the birth of many babies, it is invasive, expensive, and often does not address the proximate cause of infertility, which, for an increasing number of couples includes poor semen quality [8].

Spermatogenesis is a delicate process, particularly in humans who have poor sperm production compared to other mammals, due to a low gonadosomatic index [5]. A 59.3% decrease in average sperm counts between 1973 and 2018 has been reported in Western countries [8]. In parallel, there has been an increase in the proportion of males who are overweight or obese, both of which are strongly associated with impaired semen quality and increased sperm DNA fragmentation [16]. The relationship between decreased semen quality and obesity is thought to be associated with endocrine dysregulation, inflammation, and lipid and glucose metabolism [17]. High-fat diets have been associated with changes in the gut and seminal microbiomes, which have been found to negatively impact spermatogenesis and sperm quality and therefore male fertility [18,19,20]. As the mechanism for this is still unclear, this review summarises current research on the impact of diet and the microbiome on male infertility.

## 2. Impact of the Male Genital Tract Microbiome on Infertility

The microbiome refers to the population of microorganisms that reside in specialised niches of the body, including the skin, gut, oral cavity, vagina, and, more recently discovered, the male genital tract [21,22]. The composition and stability of the microbiome at these sites have been associated with health and disease [23]. This is an emerging field of research, with increasing evidence to support an association between lifestyle factors, such as diet and obesity, and an altered microbiome and infertility.

### 2.1. Male Genital Tract Infections

Genital tract infections, many of which are asymptomatic and as such may be better referred to as colonisations, account for approximately 15–20% of male infertility cases [24,25,26,27]. *Chlamydia trachomatis*, *Mycoplasma hominis*, *Mycoplasma genitalium* and *Ureaplasma urealyticum* are the most frequent causative agents and are treatable with antibiotics [25,28]. These infections may be acute or chronic, are often sexually transmitted, and can be asymptomatic. They can cause direct damage to the testes or seminiferous tubules, or indirectly induce inflammation leading to tissue damage, scarring or obstruction [25,29]. Spermatogenesis is thought to be impacted through direct sperm–pathogen interactions, causing agglutinations, and increased production of reactive oxygen species (ROS) and cytokines [25,30].

ROS play a vital physiological role in sperm maturation and function, supporting capacitation, the acrosome reaction, hyperactivation of the sperm tail and sperm–oocyte binding [31]. However, when balance within the antioxidant defence system is compromised, such as in the case of infection or inflammation, the production of ROS may be increased up to 100-fold, leading to the pathological effects of oxidative stress [32,33]. Oxidative damage occurs in two ways. Firstly, sperm membranes are composed of high amounts of polyunsaturated fatty acids which are susceptible to damage by ROS through lipid peroxidation [25]. Secondly, ROS can directly damage sperm DNA, leading to apoptosis of mature spermatozoa [34,35].

### 2.2. Penile and Urinary-Tract Microbiome

The male genital tract microbiome and its connection to health outcomes is an emerging area of research. A recent study suggested that the penile microbiome is relatively stable over a 12-month period with the most abundant microbes including *Corynebacterium* (16.2%), *Anaerococcus* (9.4%), *Streptococcus* (8.5%), *Finegoldia* (8.0%) and *Lactobacillus iners* (7.4%) [22]. The study identified seven different community state types influenced by circumcision status, Herpes simplex virus-2 (HSV-2) status, sexual practices, female partner bacterial vaginosis (BV) status and vaginal community state type [22]. This study identified the fact that a penile microbiome with low diversity and *Corynebacterium*-dominance may be the optimal community type for men and female partners with regard to reproductive health [22]. Conversely, a non-optimal penile microbiome was characterised by increased alpha diversity due to its association with higher prevalence of HSV-2 or with female partners with BV and non-optimal vaginal microbiome diversity [22]. These findings are consistent with other studies of the penile microbiome [36,37].

Additionally, Prodger et al. [38] found that the uncircumcised penile microbiome was associated with bacteria that increased risk of HIV infection and the attraction of HIV target cells to the inner foreskin, with *Prevotella anaerobius*, *Dialister micraerophilus*, and *Prevotella bivia* consistently appearing together in uncircumcised participants. Circumcision is associated with change in the surface microbiota, with aerobic bacteria tending to dominate post-circumcision. Whether it is the circumcision itself or the shift in the bacterial profile that explains the reduced risk of contracting HIV through heterosexual intercourse post-circumcision is unknown [38,39]. Furthermore, Osadchiy et al. [40] found that an increased relative abundance of *D. micraerophilus* in the urinary microbiome was associated with abnormal sperm motility when compared to men with normal sperm motility. Mehta et al. [23] suggested that the female reproductive tract may be the origin of the penile and urinary microbiomes, given the relative abundance of *L. iners*, a common vaginal microbiome associated with an increased risk of preterm birth and BV [41,42]. This emerging association between the penile and urinary microbiomes and sperm quality suggests an impact of these microenvironments on upstream niches, and vice versa.

### 2.3. Testicular and Seminal Microbiomes

The testes were thought to be a sterile environment until Alfano et al. [43] used next-generation sequencing (NGS), 16S rRNA gene sequencing, and identified DNA from Actinobacteria, Bacteroidetes, Firmicutes and Proteobacteria in the testicular tissue of normozoospermic men undergoing orchiectomy for testicular cancer. The study also retrieved tissue from patients with idiopathic non-obstructive azoospermia and found they lacked DNA from Bacteroidetes and Proteobacteria compared to the normozoospermic men; this gave insight into a potential variation in microbial composition between azoospermic (*n* = 5) and normozoospermic men (*n* = 5). We do note, however, that the authors acknowledged that the utilisation of non-neoplastic tissue from the normozoospermic participants who had testicular cancer may have influenced the microenvironment compared to the azoospermic participants, and no strategy for avoiding contamination of the tissue by any equipment or personnel was mentioned [43].

There is an increasing body of research employing NGS-based techniques to comprehensively characterise the seminal microbiome (Table 1), including documented associations between specific seminal microbes and semen quality. Osadchiy et al. [44] found the most abundant species in semen of men attending initial fertility assessments or men with proven paternity prior to vasectomy consultation (*n* = 73) were *Enterococcus faecalis*, *Corynebacterium tuberculostearicum*, *L. iners*, *Staphylococcus epidermis* and *Finegoldia magna*. This study found that men with abnormal sperm motility (*n* = 27) had semen with higher relative abundance of *L.iners* compared to men with normal sperm motility (*n* = 46) (9.4% vs. 2.6%, *p* = 0.046); the bacteria previously mentioned for its known association with vaginal dysbiosis [44]. Additionally, Arbelaez et al. [45] compared pre- and post-vasectomy seminal microbiomes and noted a post-vasectomy decrease in alpha diversity; this suggests that testes contribute to the microbial composition of semen or potentially more unprotected intercourse occurred post-vasectomy. Conversely, Yao et al. [46] found that *Lactobacillus*-rich seminal microbiomes were more likely to have normal leukocyte count, while Monteiro et al. [47] found that men with abnormal semen parameters had a reduction in *Lactobacillus* spp. The absence of species-level identification in these studies, however, greatly impacts interpretation of these results, considering the well-documented impacts of each species *Lactobacillus* spp. has on the vaginal microbiome [48]. Other studies have found increased seminal Firmicutes and Bacteroidetes DNA to be negatively associated with male fertility; however, given the substantial number of genera and species that these phyla constitute, such associations are of limited clinical utility and highlight a consistent limitation of many seminal microbiome studies, that of lacking taxonomic depth [49,50,51]. It is worth noting that many of these studies do not consider the cross-contamination between the semen sample with the urinary tract and the glans penis (see contamination control in Table 1). As more scientists employ full-length 16-S rRNA gene sequencing and shotgun metagenomics in seminal microbiome studies, more accurate overviews of the seminal microbiome will be attained, allowing for more clinically relevant comparison to be made.

## 3. Dietary Intake—A Potential Modifier of the Male Genital Tract Microbiome

The gut and vaginal microbiomes have both shown associations with dietary intake, which has led researchers to begin investigating a similar link with the male genital tract microbiome [56,57]. In this section, we review the current data associating diet and the seminal microbiome.

### 3.1. BMI

Body mass index (BMI) measures excess weight for an individual’s height, and is the most widely used metric for studies on overweight/obesity. However, given that BMI does not account for muscle weighing more than fat, it is a potentially inaccurate measure of overweight/obesity, especially in men. Both high and low BMI have been associated with semen-quality parameters [58,59].

Obesity is known to decrease sperm concentration, motility and semen volume and increase sperm DNA damage through several mechanisms including oxidative stress, decreased testosterone and increased scrotal temperature, due to the accumulation of inner-thigh fat [28,59]. A total of 66.5% of Australian men aged 25–34 years are currently overweight or obese [60]. Andersen et al. [61] performed semen analysis on samples from Norwegian men (*n* = 166) and found that, compared to men with a BMI between 25 and 29.9 kg/m^2^, those with an increased BMI < 35 kg/m^2^ had decreased total sperm count (203 vs. 121 × 10^6^ sperm/mL, *p* < 0.008) and progressive sperm motility (48% vs. 33%, *p* < 0.002). The high-BMI men in this study all had semen parameters close to the lower reference limit for total sperm count (39 × 10^6^/mL) and progressive sperm motility (32%), according to the *WHO Laboratory manual for examination of semen* [62].

In addition, Aminuddin et al. [63] found that obese men (*n* = 149) had lower testosterone than normal-weight men (*n* = 154) (16.18 ± 6.16 vs. 21.13 ± 6.44 nmol/L, *p* < 0.01).

Effective spermatogenesis depends on high intratesticular testosterone concentration, driven by LH-mediated Leydig cell steroidogenesis and testosterone concentration in the seminiferous tubules by androgen-binding protein, along with FSH stimulation of Sertoli cells [64]. Disruption of this endocrine process will impair spermatogenesis, semen quality and male fertility [64].

Low BMI has also been associated with poor semen quality. Ma et al. [65] conducted a study on 3966 Chinese sperm donors and found that underweight men (BMI < 18 kg/m^2^, *n* = 222) had statistically significantly lower sperm concentration (60 vs. 62 × 10^6^/mL, *p* < 0.05) and total motile sperm count (101.2 vs. 108.0 × 10^6^, *p* < 0.05), compared to the normal-weight men (BMI 18.5–24.9 kg/m^2^, *n* = 3056), though the clinical significance of such a small reduction is questionable. There are several potential causes for this slight reduction in semen quality, including malnutrition, suppression of the HPG function, or other lifestyle factors.

### 3.2. Gut Microbiome

The disruption or ‘dysbiosis’ of the gut microbiome contributes to the pathogenesis of many diseases, including autoimmune and inflammatory conditions such as allergies and inflammatory bowel disease [66,67]. The Mediterranean diet, which consists of fibre-rich foods, plus reduced quantities of red meat, saturated and trans fatty acids, is associated with increased gut microbiome diversity, decreased risk of cardiovascular disease, age-related cognitive decline, lower incidence of neurodegenerative diseases and improved semen quality [68,69,70].

Animal models suggest that a gut microbiome associated with obesity may impact spermatogenesis [4]. Microbiota transplantation from high-fat diet-induced obese mice to standard diet mice not only leads to the normal mice becoming obese, but also to localised inflammation observed at the epididymis, leading to abnormal spermatogenesis [17,71]. Furthermore, in such animal models, greater than 15% abundance of *Prevotella copri* in the gut was negatively associated with sperm motility (*n* = 20, r = −0.56, *p* = 0.011) in the high-fat-diet group [17]. This study suggests this imbalance of the gut microbial population may contribute to an increase in blood endotoxin levels which then affect spermatogenesis and, in turn, sperm motility [17]. Zhang et al. [72] reported mice with faecal microbiota transplantation-induced metabolic syndrome had a significantly reduced abundance of Ruminococcaceae_NK4A214_group associated with a reduction in bile acid levels and abnormal vitamin A metabolism, compared with non-transplanted mice. The reduction in circulating vitamin A, a pre-cursor for retinoic acid, was thought to negatively impact spermatogenesis, characterised by thinning of the murine seminiferous tubules and migration of the Sertoli cells away from the tubule basement membrane, while Leydig cells remained in contact [72]. Together, this highlights the potential connection between the gut microbiome and systemic inflammation which may impact spermatogenesis and male infertility.

### 3.3. Antioxidants

Fruit, vegetables, and nuts have been shown to improve semen quality parameters, likely due to their antioxidant properties which are known to reduce oxidative stress [68,69]. Studies have shown that high intake of vitamin C and E, beta-carotene and lycopene improve sperm quality, including concentration, motility, and morphology [73,74]. Sperm DNA damage is known to increase with age, however; Schmid et al. [75] adjusted their results by age of participant (*n* = 80) and showed that older men (>44 years) who consumed higher levels of vitamins C and E had similar levels of sperm DNA damage as young men (<44 years). Asthenozoospermic men have been found to have lower levels of co-enzyme Q10 (CoQ10), an antioxidant found naturally in the body but also in various foods [49,76]. Supplementation with CoQ10 (*n* = 35) compared to selenium (active control; *n* = 25) in idiopathic oligoasthenoteratozoospermic Iranian men was shown to increase many semen parameters, including sperm concentration (8.22 ± 6.88 vs. 12.53 ± 8.11 × 10^6^/mL, *p* < 0.01), progressive motility (16.54 ± 9.26 vs. 22.58 ± 10.15 %, *p* < 0.01) and total motility (25.68 ± 6.41 vs. 29.96 ± 8.09%, *p* < 0.01) and concentrations of enzymes involved in neutralising ROS, including serum total antioxidant capacity (1.1 ± 0.3 vs. 1.28 ± 0.26 nmol/L, *p* < 0.01), superoxide dismutase (12.6 ± 3.71 vs. 15.4 ± 4.31 U/mL, *p* < 0.01) and catalase activity (11.3 ± 2.53 vs. 12.5 ± 2.24 U/mL, *p* < 0.05) [77]. These results were consistent with earlier studies by Safarinejad et al. [78] and Balercia et al. [79]; however, Cakiroglu et al. [80] found no significant effects on motility after six months of CoQ10 supplementation in asthenoteratozoospermic men. A recent pilot study showed that antioxidant therapy involving multivitamins, CoQ10, omega-3, and oligo-elements for men with a history of failed in vitro fertilisation/intracytoplasmic sperm injection reduced DNA fragmentation from 25.8% to 18.0% (*p* < 0.0001) [81]. Overall, this suggests that enhanced antioxidant defence can lead to improvement in semen quality parameters, potentially due to the ability of the microenvironment to combat the detrimental influence of oxidative stress.

### 3.4. Fatty Acids

The composition of the sperm cell membrane is influenced by diet, particularly fatty acids, which are critical for proper sperm function. The proportion of polyunsaturated fatty acids in the spermatozoa membrane indicates sperm maturity or progression of spermatogenesis. Mature sperm membranes are comprised of approximately 20% polyunsaturated fatty acids, particularly docosahexaenoic acid, while immature sperm contain around 4% [82,83,84,85]. Humans do not synthesise polyunsaturated fatty acids, so dietary intake through nuts, seeds, and vegetable oils is essential, and consumption of such has been associated with improved semen quality [86,87]. Eslamian et al. [88] compared food-frequency questionnaire results of normozoospermic (*n* = 235) and asthenozoospermic (*n* = 107) Iranian men, and reported that men with the highest proportion of total saturated fatty acid consumption, specifically palmitic acid, and stearic acids, had increased odds of asthenozoospermia—by 2.13 and 1.9 times, respectively. Conversely, the study found that polyunsaturated fatty acids, specifically docosahexaenoic acid, reduced the risk of asthenozoospermia by 0.53 times [88]. This finding is significant, given the widespread consumption of saturated and trans fatty acids in processed food in modern Western diets, as well as the positive health associations related to adherence to the Mediterranean diet, which has higher polyunsaturated and monounsaturated fatty acid levels than the Western diet [68]. This highlights the fact that dietary intervention could be an important avenue for research in male fertility.

Interestingly, a recent meta-analysis suggested that men who had semen parameters evaluated pre and post bariatric surgery showed a normalisation of the hormone profile; however, no there was no significant improvement in semen volume, concentration, total count, morphology, total motility, progressive motility, semen pH or leukocyte numbers 6–24 months post-surgery [89]. Consistent with this, Crisóstomo et al. [90] showed, by using a mouse model, that there may be irreversible damage to testicular cells from a high-fat diet consumption. Mice fed a high-fat diet for 60 days before being switched to a standard diet for another 60 days were found to lose weight and to weigh the same as control mice at the end of the study; however, motile sperm and sperm viability did not recover compared with the control mice (35 ± 2 vs. 48 ± 2% and 35 + −2 vs. 48 ± 2%, respectively) [90]. Sixty days in the lifespan of a mouse is equivalent to approximately 18 years in humans; however, the study does not explicitly say what age the mice were when the study started [90]. This emphasises the potential importance of dietary habits in the early years of life, especially for young men, many of whom will not be considering the long-term effects of their diet on their reproductive health (Figure 1).

## 4. Limitations of Previous Research

Previous studies examining the seminal microbiome and semen quality have been limited by small sample sizes reducing the robustness of results, lack of taxonomical depth, contamination from the urinary tract, amplification of non-viable microbial DNA, and the low biomass nature of the testes. Future studies should utilise full-length 16S rRNA gene sequencing to increase taxonomic resolution whilst including appropriate controls for contamination, and considering the importance of understanding species-specific differences within microbiome niches, as is highlighted by recent studies of the vaginal microbiome [41]. Shotgun metagenomics and metabolomic/proteomic analyses may be used to explore the genetic, functional, and metabolic dynamics of the microbial communities. The utilisation of propidium monoazide (PMA), which selectively excludes the DNA of non-viable cells from amplification and subsequent analysis, may also be utilised to enhance clinical relevance of the microbiome profiles [91]. However, utilising PMA also presents its own challenges, such as variable efficiency across cell types, potential interference with PCR and the requirement of careful optimisation of experimental conditions [91]. Due to the low biomass nature of the testes, the contribution of contaminating microbial DNA associated with DNA extraction and PCR reagents also needs to be considered. This should include the use of double-strand specific DNases for pre-treatment of PCR reagents and multiple negative DNA-extraction controls that are set up in a way to account for both contaminants present within the DNA extraction reagents, as well as consumables used in sample collection [92]. Paired urine, semen, pre-washed glans penis swab, and totally negative control samples would allow for more accurate detection of the origin of microbes throughout the male genital tract.

Future studies should also consider collecting partner samples, since the vaginal microbiome is known to interact with the male reproductive tract microbiome [93,94]. There are several differences between the vaginal and male genital tract microbiomes. For example, the vagina, in the majority of women, is more acidic, due to the dominance of *Lactobacillus* spp. and the associated lower diversity, compared to the high-pH and higher-diversity microenvironment of the male genital tract microbiome [55,95]. It is known that sexual debut impacts both the vaginal and male genital tract microbiomes, with both increasing in diversity post exposure to another microbiome [96,97]. Studies have also shown that this is not just a transient exchange of microbiota, but that rather, a cohabitation of microbes has been observed, especially in people with long-term sexual partners [55]. It is hard to know exactly which environment seeds the other; however, it is known that microbes are transmitted sexually. Given the understanding of the vaginal microbiome and the impact on sexual- and reproductive-health outcomes for women, pregnancy, and neonates, the less extensive research on the male genital microbiome is currently a major barrier to optimising health outcomes for both men and women, especially when assisted reproduction is required.

It is known that BV is associated with vaginal dysbiosis, generally the depletion of *Lactobacillus*, increased anaerobe growth and increased vaginal pH [98]. The current recommended treatment for BV is oral and/or vaginal antibiotics, but this has a reoccurrence rate of >50% within 12 months. A recent study investigated an alternative, partner-inclusive treatment protocol for BV, whereby the male partner received a multisite treatment—oral and topical antibiotics in addition to the recommended treatment for the woman [99]. The study reported only a 17% reoccurrence rate, which was lower than expected and suggested it may be a strategy for a more sustained BV cure [99]. Hence, a better understanding of the male genital tract microbiome may not only address the significant gap in knowledge relating to male-factor infertility cases and associated health outcomes, but also improve the understanding and management of women’s reproductive and sexual health outcomes, as well.

## 5. Conclusions

While significant progress has been made to better understand the testes’ microenvironment, there are still significant gaps in knowledge surrounding the causes of male-factor infertility. There have been no human studies which have investigated the combination of semen quality, the seminal microbiome, and diet, to examine potential associations between the three components. If there is a modifiable lifestyle factor which is impacting male infertility, this could lead to dietary and educational interventions which could improve semen quality and reduce the need for ART. A comprehensive and accurate understanding of the composition of the male genital tract microbiome using more accurate sequencing methods, matched semen, urine, and penile samples, as well as vaginal swabs of partners, and considering impacts of lifestyle factors such as dietary intake, will give greater insight into the male genital tract microbiome and how it may be impacting fertility, and provide a more holistic view of reproductive healthcare for men and women.

## Figures and Tables

**Figure 1 microorganisms-13-00147-f001:**
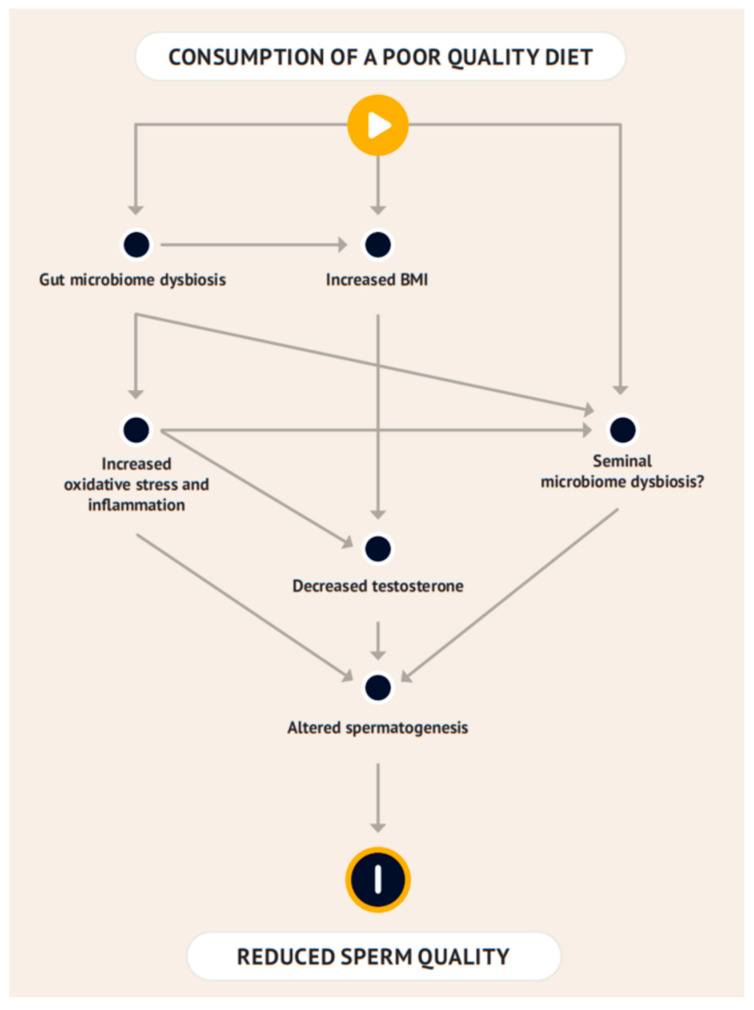
Potential interaction between poor-quality diet, the gut microbiome, seminal microbiome, and reduced sperm quality. Poor-quality diet is characterised by high intake of fats and low intake of high-resistant-starch fibres and antioxidant-rich foods.

**Table 1 microorganisms-13-00147-t001:** Summary of studies that used 16S rRNA gene sequencing to characterise the seminal microbiome.

Study	Racial/ Demographic Focus	Sample Size	Sample Type	16S Hypervariable Region	Contamination Control	Key Findings
Osadchiy et al. [44]	USA	Normal semen analysis (*n* = 42) Abnormal semen analysis (*n* = 31)	Semen	V1–V2 region	None	*Enterococcus faecalis*, *Corynebacterium tuberculostearicum*, *Lactobacillus iners*, *Staphylococcus epidermidis* and *Finegoldia magna* were the most abundant seminal microbes. *L. iners* was associated with men with abnormal sperm motility. Men with abnormal sperm concentration had higher abundance of *Pseudomonas stutzeri* (2.1% vs. 1.0% *p* = 0.024) and *P. fluroescens* (0.9% vs. 0.7%, *p* = 0.010) and lower *P. putida* (0.5% vs. 0.8%, *p* = 0.020).
Arbelaez et al. [45]	USA	Normozoospermic (*n* = 22) Pre-vasectomy (*n* = 18) Post-vasectomy (*n* = 18)	Semen	V1–V2 Region	None	A significant difference in alpha-diversity was reported between pre-vasectomy and post-vasectomy samples (*p* = 0.007). Sphingomonas, Brevundimonas and Paracoccus abundance decreased post vasectomy, while Corynebacterium increased.
Lundy et al. [21]	USA	Primary idiopathic infertility (*n* = 25) Men with proven paternity (controls, *n* = 12)	Semen, mid-stream urine, and rectal swabs	V3–V4 Region	Mid-stream urine samples were used to identify cross-contamination with semen samples	Infertile men had increased alpha-diversity, increased abundance of seminal Aerococcus and decreased Collinsella. Prevotella was positively associated with BMI and negatively associated with sperm concentration; Pseudomonas was positively associated with motile sperm count and negatively associated with semen pH.
Chen et al. [52]	Chinese	Fertile donors (control, *n* = 5) Obstructive azoospermia (*n* = 6) Non-obstructive azoospermia (*n* = 6)	Semen	V4 region	None	398 common OTUs were identified and 27 belonged to Lactobacillus. Obstructive and non-obstructive azoospermic patients had an increased abundance of Bacteroidetes and Firmicutes, while the control group had decreased Proteobacteria and Actinobacteria.
Monteiro et al. [47]	Portuguese	Fertile controls (*n* = 29) Infertile cases (*n* = 89)	Semen	V3–V6 region	Instructed to wash the glans of the penis with warm soapy water and urinate before masturbation.	Higher prevalence of pathogenic bacteria such as Neisseria, Klebsiella and Pseudomonas, and a reduction in Lactobacillus in men with hyperviscosity and oligoasthenoteratzoospermia.
Yao et al. [46]	Chinese	Normozoospermia (controls, *n* = 20) Asthenozoospermia with normal leukocyte count (*n* = 20) Leukocytospermia with normal semen parameters (*n* = 22) Leukocytospermia and asthenozoospermia (*n* = 32)	Semen	V3-V4 region	None	Firmicutes, Proteobacteria, Actinobacteria and Bacteroidetes were the most common phyla. Lactobacillus-rich microbiomes were more likely to have normal seminal leukocyte count, whereas Streptococcus-rich were more likely to have leukocytospermia.
Okwelogu et al. [50]	Nigerian	Couples (male and female; *n* = 36)	Semen Vaginal swab	V4 region	None	Species diversity was higher in seminal fluid versus the vaginal swabs. Lactobacillus was the most abundant microbe in men with normal semen parameters, followed by Gardnerella. Couples with positive IVF were significantly colonised by *Lactobacillus jensenii* and Faecalibacterium and less colonised by Proteobacteria, Prevotella, and Bacteroides, with lower Firmicutes/Bacteroidetes ratios compared to negative IVF.
Štšepetova et al. [51]	Estonian	Couples (male and female; *n* = 50)	Semen	V2–V3 region	Instructed to wash the glans of the penis with warm soapy water and urinate before masturbation.	Lactobacillus, Incertae sedis XI, Staphylococcus and Prevotella were the most common genera in semen samples. Staphylococcus was only found in patients with inflammation. Bacteroidetes had a negative correlation with sperm motility (r = −0.52, *p* < 0.05).
Baud et al. [53]	Swiss	Normal semen parameters (controls, *n* = 26) At least 1 abnormal parameter (*n* = 68)	Semen	V1–V2 region	None	Prevotella relative abundance was increased in samples with abnormal sperm motility, Staphylococcus was increased in the control group, and Lactobacillus was increased in samples with normal morphology.
Mändar et al. [54]	Estonian	Prostatitis positive (*n* = 21) Prostatitis negative (*n* = 46)	Semen, first catch urine	V6 region	First-void urine samples were used as a control to identify cross-contamination with semen samples.	Men with prostatitis had an increased species diversity compared with healthy men. Most abundant genera in all samples were Lactobacillus, Prevotella, Corynebacterium and Gardnerella. One third of first-void urine microbes were detected in semen samples.
Mändar et al. [55]	Estonian	Couples (male and female; *n* = 23)	Semen, vaginal swabs	V6 region	None	Semen samples were more diverse but had less bacterial concentration than vaginal samples, with 85% similarity between the communities. Semen samples were composed of Firmicutes, Bacteroidetes, Actinobacteria and Proteobacteria.
Weng et al. [4]	Chinese	Men from couples with infertility (*n* = 96)	Semen	V4 region	None	Lactobacillus, Pseudomonas, Prevotella and Gardnerella were the most abundant genera among samples. The majority of normozoospermic samples were Lactobacillus-rich.

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
