# Peer review of "Lower Semen Quality Among Men in the Modern Era—Is There a Role for Diet and the Microbiome?"

_microorganisms, 2025, doi:10.3390/microorganisms13010147_

Round 1

Reviewer 1 Report

Comments and Suggestions for Authors

Summary: The manuscript is a well-written and thoroughly researched contribution to a better understanding of the causes of lower semen quality among men in the modern era. The authors have effectively addressed the research question of interest, provided clear and logical arguments, and supported their claims with relevant and up-to-date citations.   

General comments:                                                                                             

There are some typographical errors and formatting adjustments noted in the manuscript text that the authors would need to address. Authors are advised to go through the manuscript thoroughly and address these errors and issues.

Specific comments:

Keyword: Authors should increase the number of keywords to at least 5 and possibly add words such as infertility, etc.

Lines 46-49: The statement made here and the context has global implications and, therefore, warrant multiple citations to support the statement.

Lines 103-114: This section seems to be supported by only a single reference. However, while this reference may provide valuable insights, the complexity of the topic warrants a more comprehensive review of the literature. Citing additional relevant studies would strengthen the argument and provide a broader context, thereby helping to validate the conclusions.

Line 192: Do not start a statement with a figure: “66.5% of Australian men aged 25-34 years are currently overweight or obese (49).”

Lines 201-202: Reference required for the …WHO Laboratory manual…referenced here.

Lines 298-99: There seems to be a missing link in the statement. Authors should consider rephrasing to enhance clarity and better understanding by the reading audience.

Reviewer 2 Report

Comments and Suggestions for Authors

The present review highlights the potential role of diet and seminal microbiome in declining of the semen quality in the modern era. The topic is very interesting and has useful information; However, it lacks demonstrative graphs illustrating the mechanistic actions of this potential role. In addition, the topic is too large to be covered in one review as each factor may need a separate review. the following items are minor comments should be addressed before acceptance.

Line 18-19: merge the two sentences with a semicolon.

Line 20: change “IN” to “In”.

Line 21: there are no citations in the abstract section.

Line 73-75: add a reference.

Line 131: remove “the”

Table 1: I think it will be better to remove the first column containing the study authorship and insert a column after the last one with the number of the reference only. If the reader needs to know more details about the study, he can check the bibliography.

Please add a paragraph to interpret the outcomes of table 1, including the effect of territory, and the washing of glans penis before semen collection or even the cross contamination between urine and semen. Do these factors make a difference in the seminal microbes.

-          The review lacks demonstrative figures highlighting the mechanistic actions controlling the pathogenesis of diet and microbiome effects on semen quality.

-          Line 248: I think there are more studies examining the effects of antioxidants dietary supplementation on semen quality.

-           
